# Post-Bariatric Hypoglycemia in Individuals with Obesity and Type 2 Diabetes after Laparoscopic Roux-en-Y Gastric Bypass: A Prospective Cohort Study

**DOI:** 10.3390/biomedicines12081671

**Published:** 2024-07-26

**Authors:** Dimitrios Kehagias, Charalampos Lampropoulos, Sotirios-Spyridon Vamvakas, Eirini Kehagia, Neoklis Georgopoulos, Ioannis Kehagias

**Affiliations:** 1Department of Medicine, University of Patras, 26504 Patras, Greece; dimikech@gmail.com (D.K.); up1084196@ac.upatras.gr (E.K.); neoklisgeorgo@gmail.com (N.G.); 2Intensive Care Unit, Saint Andrew’s General Hospital, 26504 Patras, Greece; x_lamp@hotmail.com; 3Department of Nutritional Science & Dietetics, School of Health Sciences, University of Peloponnese, 24100 Kalamata, Greece; sotvam74@gmail.com

**Keywords:** bariatric surgery, post-bariatric hypoglycemia, late dumping, diabetes mellitus, type 2, Roux-en-Y gastric bypass

## Abstract

Post-bariatric hypoglycemia (PBH) is an increasingly recognized complication after metabolic bariatric surgery (MBS). The aim of this study is to investigate potential factors associated with PBH. A cohort of 24 patients with type 2 diabetes mellitus (T2DM) and body mass index (BMI) ≥40 kg/m^2^ who underwent laparoscopic Roux-en-Y gastric bypass (LRYGBP) was retrospectively investigated for PBH at 12 months. PBH was defined as postprandial glucose at 120 min below 60 mg/dL. Questionnaires based on the Edinburgh hypoglycemia scale were filled out by the participants. Glycemic parameters and gastrointestinal (GI) hormones were also investigated. Based on the questionnaires, five patients presented more than four symptoms that were highly indicative of PBH at 12 months. According to glucose values at 120 min, one patient experienced PBH at 6 months and four patients experienced it at 12 months. Postprandial insulin values at 30 min and 6 months seem to be a strong predictor for PBH (*p* < 0.001). GLP-1 and glucagon values were not significantly associated with PBH. PBH can affect patients with T2DM after MBS, reaching the edge of hypoglycemia. Postprandial insulin levels at 30 min and 6 months might predict the occurrence of PBH at 12 months, but this requires further validation with a larger sample size.

## 1. Introduction

Post-bariatric hypoglycemia (PBH) is defined as postprandial hyper-insulinemic hypoglycemia, which occurs 2–4 h after meal consumption, presenting with Whipple’s triad criteria [1]. Since the implementation of metabolic bariatric surgery (MBS) unveiled the underlying mechanisms of hypoglycemia, the term PBH superseded the age-old term of late dumping syndrome [2]. The true prevalence of PBH is still unknown, fluctuating in the literature from 0.1 to 25%, due to the variation in diagnostic methods and hypoglycemia unawareness, which challenges the interpretation of symptoms. PBH is mainly associated with laparoscopic Roux-en-Y gastric bypass (LRYGBP), but it is also reported after sleeve gastrectomy (SG) and one-anastomosis gastric bypass (OAGB) [3]. With the advent of PBH, clinicians have been grappling with providing a common language, and, eventually, the American Society of Metabolic and Bariatric Surgery in 2017 and the European Society of Endocrinology in 2024 reached a consensus and suggested guidelines for the management of this complication. Hence, there are still severe drawbacks, such as the exact diagnostic criteria, the cut-off values for hypoglycemia, the type of provocative test, and the lack of diagnostic tools for evaluating hypoglycemia symptoms, leading to a non-standardized approach for PBH [4,5,6].

Metabolic bariatric surgery (MBS) is currently acknowledged as the most efficient method for achieving not only sustainable weight loss but also optimal glycemic control, radically changing the landscape and establishing the contemporary term of metabolic surgery [7]. The intriguing implicated pathophysiological alterations, after the changes in the gastrointestinal tract, have paved the way towards relentless research in order to elucidate the underlying mechanisms, which might serve as potential pharmaceutical targets. The gastric fundus, foregut–hindgut theory, gut microbiota, and bile acids are only some of these recognized mechanisms, while microRNAs have appeared recently as another potential mechanism [4,8,9]. Notwithstanding research endeavors that shed light on the realm of MBS, its increased prevalence has led clinicians to encounter more not-well-recognized complications, preserving the vicious cycle of obscurity and uncertainty.

Delving deeper into the pathophysiology of PBH, the exact mechanisms are still ambiguous, while it is an even more intriguing fact that patients with obesity and type 2 diabetes mellitus (T2DM) might develop postoperative hypoglycemia, reaching the opposite edge of hyperglycemia [10]. In the past, nesidioblastosis and the increased calculated β-cell area were considered the main culprits of PBH; however, partial pancreatectomies did not resolve this condition, excluding this potential mechanism [11,12]. Currently, incretins appear to have a pivotal role through positive feedback in β-cells, resulting in insulin hypersecretion, while the disruption of other counter-regulatory mechanisms such as glucagon might also be implicated in PBH occurrence [13,14].

Although PBH might be considered a rare complication, it is probably underreported, and, undoubtedly, severe PBH can have detrimental effects on quality of life, impairing cognitive function and increasing mortality. Therefore, elucidating the underlying mechanisms and implementing a standardized approach that will provide a more accurate diagnosis are mandatory necessities that will serve as a backbone for the development of targeted pharmaceutical agents, thereby providing more efficient treatment and prediction [5]. Findings from a previous randomized trial among patients with obesity and T2DM who underwent LRYGBP with or without gastric fundus resection suggested that fundus resection is not implicated in glycemic control, which was the primary outcome of the study [8]. Among the studied population, especially one year postoperatively, profound PBH was observed in some patients 2 h after the oral glucose tolerance test (OGTT), and they manifested neuroglycopenic symptoms. The aim of this study is to investigate this cohort in terms of factors associated with PBH in patients with obesity and T2DM and suggest mechanisms that might be implicated in this increasingly recognized complication.

## 2. Materials and Methods

### 2.1. Study Design

This is an observational cohort study with prospectively collected samples and a retrospective interview of the patients, which was conducted in the bariatric unit of the University Hospital of Patras in Greece and included patients with obesity and T2DM. Initially, these patients were enrolled in a randomized clinical trial forming two intervention arms: LRYGBP and LRYGBP with fundus resection [8]. The trial was registered at clinicaltrials.gov (NCT05854875), and the results of this study have been previously published. The total included 24 patients who were retrospectively investigated for PBH at 6 and 12 months postoperative. These patients were selected from the registry of the metabolic and bariatric unit and they signed informed consents. All procedures performed were in accordance with the 1964 Helsinki Declaration, and this study was approved by the local research and ethics committee (No. 730/10.12.2019) [15]. This cohort study was performed following the strengthening of the criteria for reporting cohort, cross-sectional, and case–control studies in surgery (STROCSS 2021) [16]. The primary outcome was postprandial glucose levels at 120 min during OGTT 12 months postoperatively. The secondary outcomes included insulin, glucagon, GLP-1 levels, calculated β-cell area, and insulinogenic index. Finally, a potential correlation of the aforementioned parameters at 6 and 12 months, with PBH at 12 months, was also investigated.

### 2.2. Patient Population

The studied cohort included 24 patients aged between 18 and 60 years old, with a BMI ≥40 kg/m^2^ and T2DM defined by the criteria of the American Diabetes Association (ADA) [17]. The patients underwent LRYGBP and LRYGBP with fundus resection, and they were evaluated preoperatively and at 6 and 12 months postoperatively. The duration of T2DM was determined to be less than 8 years, since longer durations are related to irreversible β-cell function impairment and poorer postoperative recovery. All participants were evaluated by different specialties, and behavioral and anthropometric parameters and metabolic profiles were assessed. The exclusion criteria were gestation, diabetes mellitus type I, alcohol or drug abuse, major depressive disorder, non-compliance with the medical personnel’s instructions, and previous abdominal surgeries with altered gastrointestinal anatomy.

All operations were completed laparoscopically by the same surgeon in the surgical department of the tertiary referral hospital. Both procedures included the creation of a small gastric pouch of 30 mL capacity, a very long biliopancreatic limb of 200 cm, and an alimentary limb of 150 cm. Gastrojejunal anastomosis was created with a circular stapler of 25 mm diameter after the transoral placement of the anvil with assistance from the anesthesiologist. In the fundus resection group, the gastric body and fundus were mobilized by dividing the gastrocolic ligament and short gastric vessels until exposing the angle of His. A gastric fundus with dimensions of approximately ±5.5 cm (width) and ±10 cm (vertical length) was removed with the use of a linear stapler [8].

One year after the end of the randomized clinical trial, these patients were contacted and responded to a questionnaire in order to evaluate symptoms of PBH. The questionnaire was adapted from the Edinburgh hypoglycemia scale, which consists of 11 key symptoms associated with hypoglycemia, and they are segregated into three categories: autonomic, neuroglycopenic, and malaise. Specifically, these symptoms include sweating, palpitation, shaking, hunger, confusion, drowsiness, odd behavior, speech difficulty, incoordination, nausea, and headache. This model has been externally validated and is suggested as a standardized approach for evaluating hypoglycemia symptoms [18]. Patients were considered arbitrarily highly suspicious for PBH if they exhibited more than 4 symptoms of the Edinburgh scale. The imposed question was “Did you feel any of the aforementioned symptoms 1 to 3 h postprandially?” The onset of symptoms and the random self-measurement of glucose levels during these episodes were also questioned and evaluated. All patients were postoperatively instructed by expert dietitians to follow a diet enriched in protein with low carbohydrates. However, the exact type of meal consumed before the occurrence of these symptoms was not reported due to the retrospective nature of the questionnaire and the inability of patients to reliably determine this.

### 2.3. Laboratory Measurements

As described in the previous publication, blood samples were collected during a 75 gr OGTT at 0, 30, 60, and 120 min. This procedure was repeated preoperatively at 6 and 12 months. The measurement of insulin and GLP-1 levels was explicitly described previously, while these prospectively collected stored samples were also utilized for the measurement of glucagon. These samples were in ethylene-diamine-tetra-acetic acid (EDTA) vials, which contained 1.8 TIU (trypsin inhibitor units) of proteinase inhibitor, aprotinin (Trasylol). They were centrifuged at 4 °C for 20 min and at 1600 RCF (relative centrifuge force), and they were stored at −70 °C. Glucagon was measured with commercial ELISA kits (Invitrogen, ThermoFisher Scientific, catalog number EHGCG), and the minimum detectable dose was 2.5 pg/mL. The aforementioned gastrointestinal (GI) hormones were all measured during the OGTT at 0, 30, 60, and 120 min preoperatively at 6 and 12 months. Fasting c-peptide was calculated preoperatively at 6 and 12 months with commercial ELISA kits and used for estimating the “calculated β-cell area” index.

### 2.4. Definitions and Criteria

For defining hypoglycemia, a cut-off value of 60 mg/dL was selected. Based on the recommendations from the American Diabetes Association (ADA) and the European Association for the Study of Diabetes (EASD), a cut-off value of 54 mg/dL is suggested since glucose levels below this threshold are related to impaired cognitive function and neuroglycopenic and autonomic symptoms [19]. However, we selected a slightly higher cut-off value because patients with T2DM were investigated. Beta-cell function was assessed with the use of the insulinogenic index, which was calculated as the ratio of insulin change to glucose change from 0 to 30 min. The insulinogenic index mainly describes the first phase of insulin secretion, which is remarkably associated with β-cell function [20].

The postprandial values of glucose, insulin, glucagon, and GLP-1 were estimated with area under the curve (AUC) using the trapezoidal method. Supposing that GI hormones are implicated in PBH, we particularly evaluated the postprandial levels of insulin and GLP-1 at different time points during the OGTT. Specifically, based on the literature, an abrupt postprandial increase in insulin and GLP-1 is observed at 30 and 60 min postprandially. Therefore, the postprandial levels of insulin and GLP-1 at these time points were investigated for a potential correlation with PBH. Finally, the ratio of fasting c-peptide to fasting glucose (ng/mL × mg/dL) was utilized to estimate the calculated β-cell area. This index was validated by Meier et al., and it exhibited a significant linear correlation with the histological assessment of the β-cell area [21,22].

### 2.5. Statistical Analysis

For statistical analyses, the SPSS version 20.0 statistics software package (Statistical Package for the Social Sciences, SPSS Inc., Chicago, IL, USA) was utilized, while graphs were generated using GraphPad Prism version 5.0 (Graph-Pad Software, Inc., San Diego, CA, USA). For continuous parameters, the mean ± standard error (SE) was utilized. The non-parametric Kruskal–Wallis test was used for comparing parameters at different time points (baseline, 6 months, and 12 months). Statistical significance was set at p T2DM0.05 and was adjusted using the Bonferroni correction.

For the correlation of different parameters with PBH, a biserial correlation was applied, where PBH was the dichotomous variable (0 for glucose above 60 mg/dL and 1 for glucose below 60 mg/dL). For assessing predictors of PBH, a stepwise binary regression analysis was used, and the aforementioned binary variable was included for PBH. Finally, receiver operating characteristic (ROC) curves were generated to assess the reliability of predictive factors for PBH diagnosis at 12 months.

## 3. Results

### 3.1. Baseline and Follow-Up Characteristics

Eleven male and thirteen female patients were enrolled in the cohort study, with a mean age of 46.75 ± 10.98 years. Preoperatively, the mean weight and BMI were 153.15 ± 36.8 kg and 53.2 ± 10.6 kg/m^2^, respectively. Weight loss was remarkable postoperatively, and at 12 months, there was a significant decrease in terms of weight and BMI: 89.43 ± 18.6 kg (*p* < 0.05) and 30.62 ± 4.81 kg/m^2^ (*p* < 0.05), respectively. EWL% was significantly increased between 6 and 12 months from 54.55 ± 7.66 to 70.33 ± 12.4 (*p* < 0.05).

Regarding T2DM, the patients had a mean HbA1c of 7.94 ± 1.74 and c-peptide of 4.42 ± 1.89 ng/mL, with a mean preoperative duration of 3.77 ± 2.3 years. At 6 months, both HbA1c and c-peptide significantly decreased (*p* < 0.05) and, at 12 months, they remained fairly constant at 5.36 ± 0.55 and 2.65 ± 1.96, respectively. Based on glucose measurement during the OGTT, preoperatively, none of the patients presented PBH. Conversely, at 6 months, one patient (4%) experienced glucose levels below 60 mg/dL, and at 12 months, the number of these patients quadrupled, reaching 16.6% (Table 1). Regarding the preoperative diabetes treatment, only one patient was completely off anti-diabetic medications. The other twenty-three patients were on oral medications, while seven took insulin and four GLP-1 analogs. Postoperatively, 95% (23/24) showed complete diabetes remission at one year without taking any medication. Only one patient from the LRYGBP with fundus resection group experienced relapse and was re-introduced to oral medications at one year. Finally, PBH was not associated with the type of surgery since PBH occurred in patients from both groups.

### 3.2. Hypoglycemia Symptoms and Edinburgh Questionnaire

The questionnaires regarding PBH were answered retrospectively by all 24 patients enrolled in the initial randomized controlled study. Five patients presented more than four symptoms on the Edinburgh scale and were considered highly suspicious for PBH.

Overall, the most common symptoms were autonomic, specifically sweating, palpitation, and shaking comprising 25%, 33%, and 21%, respectively. Regarding neuroglycopenic symptoms, drowsiness was the most common at 12.5%, followed by confusion at 8%, while hunger, odd behavior, speech difficulty, and incoordination were equally presented at 4%. Finally, as far as malaise is concerned, nausea comprised 21% and headache comprised 4%. Sweating, palpitation, shaking, and nausea were present in all five patients, which are symptoms highly indicative of PBH (Table 2).

Regarding the time of the presentation, only one patient reported these symptoms preoperatively; three patients reported these symptoms at 6 months, and five patients reported them during evaluation (Table 1). None of the patients who participated in the questionnaire reported any syncope or a random measurement of glucose value below 60 mg/dL.

### 3.3. Glycemic Parameters and Calculated β-Cell Area

Glycemic control was achieved from the 6th month, with patients demonstrating a significant decrease in glucose and insulin fasting levels (Table 3; Figure 1). Specifically, from baseline to 12 months postoperatively, glucose fasting declined from 130.54 ± 51.15 to 88.13 ± 21.52 mg/dL (*p* < 0.05), while insulin fasting remarkably decreased from 23.91 ± 13.6 to 9.83 ± 13.36 μIU/mL (*p* < 0.05). Along with the glycemic improvement, β-cell function also witnessed an outstanding enhancement, with the insulinogenic index increasing from 0.5052 ± 0.44 at baseline to 0.9862 ± 1 at 6 months (*p* < 0.05) and 1.483 ± 1.63 at 12 months (*p* < 0.05). Considering that, at 30 and 60 min postprandially, insulin demonstrates peak secretion, these time points were evaluated and exhibited an increase in mean values, not reaching statistical significance (*p* > 0.05). Based on the graph, the first and second phases of insulin secretion were improved (Figure 1).

In order to investigate the theory of nesidioblastosis, the calculated β-cell area was evaluated 6 and 12 months postoperatively. The mean values were comparable at all time points (*p* < 0.05) without showing changes in calculated β-cell area postoperatively (Table 3).

### 3.4. GLP-1 and Glucagon

GLP-1 demonstrated a significant increase postoperatively in both fasting and postprandial levels. Specifically, fasting levels increased from 0.78 ± 0.14 to 1.39 ± 0.58 pg/mL (*p* < 0.05), while GLP-1 improved its pattern secretion with peak release at 30 min postprandially compared with preoperatively (*p* < 0.05). GLP-1 AUC levels also significantly increased from 94.72 ± 19.63 to 182.77 ± 87.19 (*p* < 0.05) (Table 4, Figure 1 and Figure 2).

Glucagon fasting and postprandial levels during OGTT did not show a significant change from baseline to 6 or 12 months (*p* > 0.05). Based on the graphs, fasting and AUC levels followed the same pattern, experiencing a slight increase at 6 months and returning to preoperative levels without achieving statistical significance at 12 months (*p* > 0.05) (Table 4, Figure 1 and Figure 2).

### 3.5. Correlations and Predictive Factors

According to the biserial correlation, postprandial insulin levels at 30 and 60 min—at 6 and 12 months postoperatively—were significantly positively related to the binary variable of glucose 120 min at 12 months (0 for glucose > 60 mg/dL and 1 for glucose < 60 mg/dL), meaning that increasing values at these time points were associated with PBH. Conversely, GLP-1 levels at these time points were not associated with PBH (Table 5).

The high correlation coefficient between the binary glucose variable at 12 months postoperatively and postprandial insulin at 30 and 60 min at 6 months postoperatively was an unexpected and interesting finding. In order to investigate whether the postprandial levels of insulin at 6 and 12 months could be a useful marker for PBH, a stepwise binary regression analysis was performed in order to determine the strongest predictive factor from the aforementioned parameters. The results of the analysis indicated that postprandial insulin levels at 30 min and 6 months might be a strong predictor for PBH at 12 months (*p* < 0.001). The rest of the tested variables did not show a similar potential and were excluded from the model (Table 6).

The final step was validating the accuracy of the postprandial insulin levels at 30 min and 6 months as a potential prognostic marker. The validation was assessed by generating ROC curves. The positive result was the occurrence of PBH. The ROC curves revealed that insulin at 30 min at 6 months was a sensitive and reliable biomarker since its calculated AUC value was the highest at 0.933 (Figure 3). The rest of the tested biomarkers showed lower values of AUC, as depicted in Figure 3. Therefore, this reinforces the potential use of insulin at 30 min and 6 months as a potential PBH prognostic biomarker.

## 4. Discussion

The aim of this study was to investigate PBH in a cohort of patients with T2DM and obesity after LRYGBP and identify potentially involved mechanisms that could serve as tools for predicting the occurrence of this complication. Postprandial insulin levels are associated with the presence of PBH; in particular, insulin levels at 30 min during OGTT at 6 months might be used as a predictor for PBH. Conversely, postprandial levels of GLP-1 and glucagon, as well as the calculated β-cell area, are not significantly related to PBH. The type of surgery is not associated with PBH, further supporting our previous findings, which showed that fundus resection is not implicated in glycemic control [8].

Although PBH has been recognized as a complication of MBS for several years, its true incidence is not yet determined. From studies utilizing continuous glucose monitoring (CGM), this incidence increases from 25 to 75%, while after provocative tests, namely the OGTT and mixed meal test, it is estimated to be between 19 and 30% [23,24]. On the basis of our findings, 16% of the patients presented postprandial glucose levels below 60 mg/dl after 12 months, and 20% had symptoms highly indicative of PBH according to the retrospective questionnaire evaluation. However, it is interesting to note that only patients with T2DM were included, which contradicts some observational studies that propose preoperative T2DM as a protective factor for PBH [23]. The mean duration of preoperative T2DM in our cohort was 3.7 years, and, most probably, the irreversible damage of β-cells had not occurred. As a result, after LRYGBP and gastrointestinal anatomy alterations, β-cell function was restored, and glycemic control was achieved. In line with this notion, Raverdy et al. suggested that PBH after LRYGBP occurs in patients with higher pre-existing beta-cell functions, as estimated via the insulinogenic index, when insulin sensitivity is restored in relation to weight loss, which more succinctly explains our findings [22]. The mean preoperative insulinogenic index of the four patients of the cohort who presented PBH was 0.654, and it showed a remarkable four-fold increase at 12 months, reaching 2.5, which is far higher than the mean value of the insulinogenic index of the cohort, as depicted in Table 3. This indicates that after, achieving weight loss and improving insulin sensitivity, β-cell function is restored, and patients with an inherent increased β-cell function will demonstrate increased values of the insulinogenic index.

Another demanding part of PBH diagnosis is the objective report of the symptoms and the diagnostic methods used. It is well known that PBH can present without any symptoms, resulting in the term “hypoglycemia unawareness”, as demonstrated mainly from studies using CGM [23,25]. Furthermore, the lack of clinical tools is a severe drawback for assessing PBH. Many studies investigating PBH have implemented the Sigstad score, which is a false approach because this score is used for detecting early dumping syndrome [26]. In line with this notion, a recently published study found that the Sigstad score is not a reliable or valid method for detecting late dumping syndrome after MBS [27]. Perhaps, the Edinburgh hypoglycemia scale test and instructions for patients with respect to measuring blood glucose levels whenever they present these symptoms would be a more rational approach.

The optimal diagnostic approach for the diagnosis of PBH has not yet been determined by clinicians. From provocative tests, the mixed meal test appears as a more preferred method because it resembles an ordinary meal. Conversely, OGTT, which was also used in this cohort, can result in PBH in 70% of RYGB patients, with small differences between symptomatic and asymptomatic patients, rendering OGTT less useful in confirming the diagnosis [5]. Although the documented glucose values in the studied cohort were after OGTT, the type of meal before the onset of symptoms was unknown when patients filled out the questionnaires. The lack of information was a major drawback because it can affect the occurrence of PBH. Specifically, meals with a high carbohydrate load are the main culprits for PBH and should be replaced with small meals enriched in protein and fiber [1]. Finally, CGM over the course of 3 days and during normal meals is more sensitive and is perhaps the most promising method [28]. Nevertheless, more evidence is required based on the recent guidelines of the European Society of Endocrinology [8].

Delving deeper into the pathophysiology, the most compelling theory revolving around PBH is the exaggerated insulinotropic response, resulting in overwhelming hypoglycemia [29]. This increased postprandial insulin secretion is attributed to elevated β-cell sensitivity during hyperglycemia, while concurrently blunted insulin suppression is witnessed when glucose falls below fasting levels [30,31,32]. In our study, postprandial insulin levels at 30 and 60 min and 6 months or 12 months were significantly associated with postprandial glucose values below 60 mg/dL. Specifically, postprandial insulin levels at 30 min at 6 months might be used as a predictive factor for PBH at 12 months, although this should be validated in a larger sample. Turning to details, the mean values of postprandial insulin levels at 30 min experienced by the four patients with PBH, at baseline, 6 months, and 12 months were 74.25, 164.82, and 209.25 μIU/mL, respectively, showing a remarkable postoperative increase when insulin sensitivity and glucose homeostasis were restored. Whether this is a result of an increase in pre-existing β-cell function that was restored after weight loss or dysregulation in β-cell response requires more investigation since many factors are implicated.

Apart from insulin’s action, its secretion is also closely related to enteroinsular axis activity, which is mediated by incretins GLP-1 and GIP. Particularly, postprandial GLP-1 levels are significantly positively associated with insulin secretion, bolstering the fact that incretin contributes to enhanced β-cell function postoperatively [3,33]. Utilizing more evidence about PBH, patients who experience hypoglycemia after RYGBP are reported to have enhanced GLP-1 responses to meal ingestion compared with asymptomatic RYGB individuals [34]. In our study, fasting and postprandial GLP-1 levels were remarkably improved at 6 and 12 months, and they were most probably the main culprits that led to optimal glycemic control and T2DM remission in the study cohort. Supposing that GLP-1 contributes to insulin secretion, we hypothesized that postprandial GLP-1 levels at 30 min, where the maximum secretion is observed, depending on the values, could be involved in PBH at 12 months. Nevertheless, GLP-1 levels at 30 min and even at 60 min could not be used, surprisingly, as predictors for PBH at 12 months based on binary regression analysis. Furthermore, from the biserial correlation, the aforementioned parameters were not significantly related to PBH, creating controversy regarding their role in PBH. Therefore, according to our findings, an inherent pre-existing β-cell function with insulin hypersecretion seems to be a prerequisite for PBH rather than GLP-1 levels alone.

Apart from incretins and insulin, the disruption of counter-regulatory mechanisms, such as glucagon, might be implicated in PBH, increasing the frequency of these episodes [35]. While glucagon is normally released in response to hypoglycemia, stimulating hepatic glucose output, after RYGBP, the postprandial glucagon response among patients with and without hypoglycemia is identical, indicating a dysregulated a-cell response to hypoglycemia [36]. Particularly in T2DM patients, the paracrine control of a-cell glucagon is jeopardized, resulting in a-cell insensitivity, increased fasting glucagon levels, and blunted postprandial glucagon suppression [37]. In our study, postprandial glucagon levels did not significantly change. However, fasting and postprandial levels were slightly decreased at 12 months compared with 6 months, showing a gradual improvement in a-cell sensitivity. Moreover, from the biserial correlation, glucagon levels at any time point were not correlated with the PBH, exhibiting a more limited role. Hence, the attenuation of counter-regulatory mechanisms in T2DM patients, together with the significant changes that occurred after RYGBP, create complex interactions that require further investigation. Finally, based on a recently published study, glucagon’s insulinotropic role is witnessed after MBS, which is mediated through the GLP-1 receptor on β-cells, further complicating glucagon action [38].

A while after the turn of the millennium, Service et al. published a landmark study in order to explain PBH, and they suggested the increased calculated β-cell area after RYGBP as the main culprit because nesidioblastosis characteristics were observed in specimens from patients undergoing partial pancreatectomy for PBH [12,39]. However, the theory of nesidioblastosis was steadily abandoned because the removal of islet cells via partial pancreatectomy did not entirely resolve hypoglycemia. On the other hand, it was observed that β-cell nuclear diameter was increased with respect to BMI, and this was preserved together with β-cell hyperfunction after RYGBP. In line with this notion, we evaluated calculated β-cell area, as indicated by Meier et al., and no significant changes were observed postoperatively. Moreover, the calculated β-cell area was not associated with PBH in the biserial correlation. Therefore, according to the literature and our findings, pre-existing β-cell hyperfunction, which persists after weight loss, is probably the key leverage point for PBH [40,41].

Although these findings can be considered intriguing, there are several major limitations, requiring a cautious interpretation of the results. First and foremost, PBH was not the primary endpoint or one of the secondary endpoints of the initially designed randomized controlled study. PBH was mainly observed at twelve months after RYGBP through patients reporting symptoms of hypoglycemia or low glucose values, and this guided us in retrospectively evaluating these patients even though the samples were prospectively collected. Regarding the provocative test used, OGTT might not be the most ideal because it overdiagnoses PBH. Furthermore, except for the small sample size of the study cohort, which severely weakens our findings, the postoperative time period of one year is also not the most appropriate, since the physiological changes after MBS stabilize after one year, enabling us to safely study the involved mechanisms. Another major limitation of this study was the lack of data regarding the type of meal these patients consumed when they completed the questionnaire; this is because the consumption of meals with low carbohydrate and high protein levels is the cornerstone for avoiding PBH. Hence, the laboratory values of glucose below 60 mg/dl were documented after OGTT for all patients. Other than these severe limitations, the study cohort only included patients with T2DM and investigated several mechanisms implicated in PBH. Despite the small sample size, the potential predictive value of postprandial insulin levels at 6 months for PBH and 12 months and the role of GLP-1 and glucagon revolving around PBH are valuable findings, providing insights in a not yet clearly elucidated field. In any case, PBH cannot be easily investigated due to hypoglycemia unawareness and the inability to objectively report patients’ symptoms.

## 5. Conclusions

According to our findings, pre-existing β-cell hyperfunction, which persists postoperatively after weight loss and might lead to increased postprandial insulin levels, is associated with PBH. Moreover, the postprandial insulin levels at 30 min and 6 months might be used as a predictive factor for PBH at 12 months, while GLP-1 and glucagon levels are not associated with PBH among patients with diabetes after LRYGBP. The predictive role of insulin can raise awareness among clinicians to adequately prepare for and inform patients to avoid the devastating effects of PBH; this should be validated using a larger sample size. Future research should aim to investigate the implicated mechanisms with well-designed studies, while clinical tools for objectively assessing PBH should also be validated.

## Figures and Tables

**Figure 1 biomedicines-12-01671-f001:**
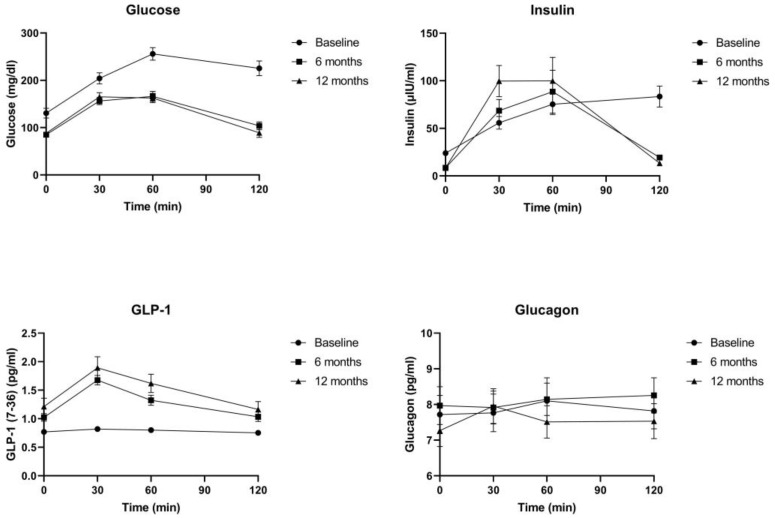
Graphs of glucose, insulin, and GI hormones during OGTT at baseline and 6 and 12 months.

**Figure 2 biomedicines-12-01671-f002:**
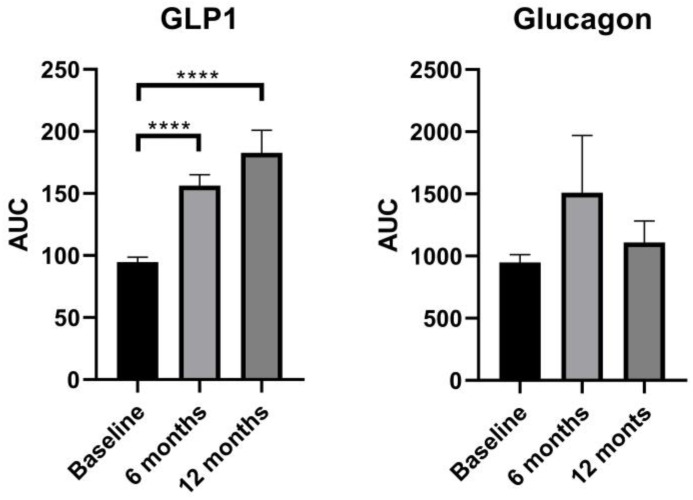
Bar graphs of GLP-1 and glucagon AUC levels at baseline and 6 and 12 months. ********
*p* < 0.0001.

**Figure 3 biomedicines-12-01671-f003:**
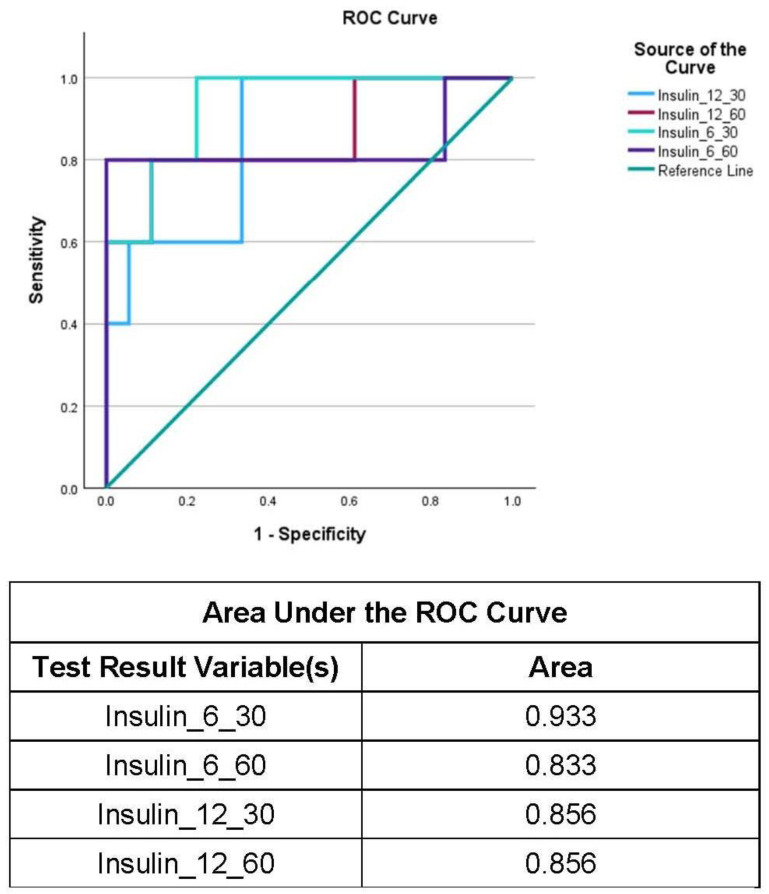
ROC curves for postprandial insulin values and AUC levels. The proposed early PBH biomarker (Insulin_6_30) shows the highest AUC value. The rest of the postprandial insulin levels tested show lower values of AUC.

**Table 1 biomedicines-12-01671-t001:** Baseline and follow-up characteristics of the patients.

Parameters	Baseline	6 Months		12 Months		
	Mean ± SD	Mean ± SD	*p* 0–6	Mean ± SD	*p* 0–12	*p* 6–12
Age (year)	46.75 ± 10.98					
Gender (M) (F)	(11) (13)					
Weight (kg)	153.15 ± 36.8	104.62 ± 23.66	**<0.05**	89.43 ± 18.6	**<0.05**	0.173
BMI (kg/m^2^)	53.2 ± 10.6	36.03 ± 6.05	**<0.05**	30.62 ± 4.81	**<0.05**	**<0.05**
EWL %		54.55 ± 7.66		70.33 ± 12.4		**<0.05**
HbA1c (%)	7.94 ± 1.74	5.31 ± 0.53	**<0.05**	5.36 ± 0.55	**<0.05**	0.909
c-peptide (ng/mL)	4.42 ± 1.89	2.77 ± 0.88	**<0.05**	2.65 ± 1.96	**<0.05**	0.503
T2DM duration (year)	3.77 ± 2.3					
Anti-diabetic medications (insulin) (GLP1 analogs)	23 (7) (4)	0 (0) (0)		1 (0) (0)		
Patients with symptoms of PBH (%)	1 (4)	3 (12.5)		5 (20.8)		
Patients with Glu at 120 min < 60 mg/dL (%)	0 (0)	1 (4)		4 (16.6)		

Values are expressed as mean ± SD; M (male); F (female); EWL (excess weight loss); T2DM (type 2 diabetes mellitus); PBH (post-bariatric hypoglycemia); *p* 0–6 (between baseline and 6 months); *p* 0–12 (between baseline and 12 months); *p* 6–12 (between 6 and 12 months); *p* < 0.05 bold for significance.

**Table 2 biomedicines-12-01671-t002:** Patients with hypoglycemia symptoms based on the questionnaires adjusted for the Edinburgh scale. High clinical suspicion for patients presenting above 4 symptoms.

Symptoms	Overall Patients (*n* = 24) *n* (%)	Patients with High Clinical Suspicion (*n* = 5) *n* (%)
Autonomic		
Sweating	6 (25)	5 (100)
Palpitation	8 (33)	5 (100)
Shaking	5 (21)	5 (100)
hunger	1 (4)	1 (20)
Neuroglycopenic		
Confusion	2 (8)	2 (40)
Drowsiness	3 (12.5)	3 (60)
Odd behavior	1 (4)	0 (0)
Speech difficulty	1 (4)	1 (20)
Incoordination	1 (4)	1 (20)
Malaise		
Nausea	5 (21)	5 (100)
headache	1 (4)	1 (20)

**Table 3 biomedicines-12-01671-t003:** Glycemic parameters and calculated β-cell area at baseline and 6 and 12 months.

Parameters	Baseline	6 Months		12 Months		
	Mean ± SD	Mean ± SD	*p* 0–6	Mean ± SD	*p* 0–12	*p* 6–12
Glu fasting (mg/dL)	130.54 ± 51.15	85.08 ± 13.79	**<0.05**	88.13 ± 21.52	**<0.05**	0.885
Glu 120 min (mg/dL)	225.29 ± 76.88	103.87 ± 39.94	**<0.05**	89.04 ± 47.74	**<0.05**	0.597
Insulin fasting (μIU/mL)	23.91 ± 13.6	8.52 ± 5.77	**<0.05**	9.83 ± 13.36	**<0.05**	0.765
Insulin 30 min (μIU/mL)	55.8 ± 32.45	68.52 ± 57.56	0.680	99.63 ± 78.66	0.103	0.263
Insulin 60 min (μIU/mL)	75.36 ± 52.64	88.53 ± 110.4	0.588	99.87 ± 118	0.834	0.744
Insulin AUC	7927.5 ± 4530	6747.6 ± 6688	0.177	8030 ± 7550	0.836	0.433
Insulinogenic index	0.5052 ± 0.44	0.9862 ± 1	**<0.05**	1.483 ± 1.63	**<0.05**	0.352
Calculated β-cell area	0.035 ± 0.01	0.032 ± 0.012	0.869	0.03 ± 0.02	0.228	0.321

Values are expressed as mean ± SD; *p* < 0.05 bold for significance.

**Table 4 biomedicines-12-01671-t004:** Gastrointestinal hormones glucagon and GLP-1 at baseline and 6 and 12 months.

Gastrointestinal Hormones	Baseline	6 Months		12 Months		
	Mean ± SD	Mean ± SD	*p* 0–6	Mean ± SD	*p* 0–12	*p* 6–12
Glucagon fasting (pg/mL)	7.71 ± 2.62	7.96 ± 2.48	0.637	7.26 ± 2.05	0.502	0.266
Glucagon AUC	948.25 ± 307.9	971.19 ± 250.52	0.406	911.68 ± 252.5	0.933	0.375
GLP-1 fasting (pg/mL)	0.78 ± 0.14	1.14 ± 0.48	**<0.05**	1.39 ± 0.58	**<0.05**	0.184
GLP-1 AUC	94.72 ± 19.63	156.25 ± 43.1	**<0.05**	182.77 ± 87.19	**<0.05**	0.650

*p* < 0.05 bold for significance.

**Table 5 biomedicines-12-01671-t005:** Biserial correlation of parameters with the binary variable of glucose at 120 min at 12 months.

		Cat_glu12_120	Insulin_6_30	Insulin_6_60	GLP1_6_30	GLP1_6_60	Insulin_12_30	Insulin_12_60	GLP1_12_30	GLP1_12_60
Cat_glu12_120	Pearson Correlation	1	0.709 **	0.600**	−0.063	−0.010	0.568**	0.694 **	0.083	−0.016
	Sig. (2-tailed)		0.000	0.002	0.775	0.962	0.005	0.000	0.706	0.944
	N	23	23	23	23	23	23	23	23	23
Insulin_6_30	Pearson Correlation	0.709 **	1	0.823 **	−0.172	−0.151	0.757 **	0.868 **	−0.070	−0.200
	Sig. (2-tailed)	0.000		0.000	0.422	0.480	0.000	0.000	0.751	0.359
	N	23	24	24	24	24	23	23	23	23
Insulin_6_60	Pearson Correlation	0.600 **	0.823 **	1	−0.210	−0.181	0.675 **	0.936 **	0.213	−0.182
	Sig. (2-tailed)	0.002	0.000		0.324	0.396	0.000	0.000	0.329	0.407
	N	23	24	24	24	24	23	23	23	23
GLP1_6_30	Pearson Correlation	−0.063	−0.172	−0.210	1	0.182	−0.085	−0.109	−0.004	0.197
	Sig. (2-tailed)	0.775	0.422	0.324		0.395	0.699	0.621	0.987	0.369
	N	23	24	24	24	24	23	23	23	23
GLP1_6_60	Pearson Correlation	−0.010	−0.151	−0.181	0.182	1	−0.163	−0.214	−0.269	0.162
	Sig. (2-tailed)	0.962	0.480	0.396	0.395		0.458	0.327	0.214	0.460
	N	23	24	24	24	24	23	23	23	23
Insulin_12_30	Pearson Correlation	0.568 **	0.757 **	0.675 **	−0.085	−0.163	1	0.827 **	−0.130	−0.335
	Sig. (2-tailed)	0.005	0.000	0.000	0.699	0.458		0.000	0.555	0.118
	N	23	23	23	23	23	23	23	23	23
Insulin_12_60	Pearson Correlation	0.694 **	0.868 **	0.936 **	−0.109	−0.214	0.827 **	1	0.125	−0.244
	Sig. (2-tailed)	0.000	0.000	0.000	0.621	0.327	0.000		0.570	0.261
	N	23	23	23	23	23	23	23	23	23
GLP1_12_30	Pearson Correlation	0.083	−0.070	0.213	−0.004	−0.269	−0.130	0.125	1	−0.075
	Sig. (2-tailed)	0.706	0.751	0.329	0.987	0.214	0.555	0.570		0.734
	N	23	23	23	23	23	23	23	23	23
GLP1_12_60	Pearson Correlation	−0.016	−0.200	−0.182	0.197	0.162	−0.335	−0.244	−0.075	1
	Sig. (2-tailed)	0.944	0.359	0.407	0.369	0.460	0.118	0.261	0.734	
	N	23	23	23	23	23	23	23	23	23

** Correlation is significant at the 0.01 level (2-tailed); cat_glu12_120 (dichotomous variable of glucose levels at 120 min at 12 months: 0 for above 60 mg/dL and 1 for below 60 mg/dL).

**Table 6 biomedicines-12-01671-t006:** Stepwise binary regression analysis and excluded variables.

Model Summary							
Model	R	R^2^	Adj R^2^	SEE	R^2^ Change	F Change	Sig. F Change ^a^
1	0.709 ^a^	0.502	0.479	0.30454	0.502	21.193	0.000
**Excluded Variables**							
**Model 1**	**Beta In**	**t**	**Sig.**	**Partial Correlation**	**Collinearity Statistics**		
					**Tolerance**		
Insulin_12_30	0.074 ^b^	0.306	0.763 ^b^	0.068	0.428		
Insulin_12_60	0.319 ^b^	1.029	0.316 ^b^	0.224	0.246		
Insulin_6_60	0.057 ^b^	0.207	0.838 ^b^	0.046	0.328		

The dependent variable is the categorical variable of postprandial glucose at 120 min and postoperative 12 months (Cat Glu12 120). Only postprandial insulin at 30 min and 6 months is calculated as a prognostic biomarker (^a^). The rest of the tested parameters have no prognostic potential (^b^).

## Data Availability

The data presented in this study are available from the corresponding author upon request.

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
