# Peer review of "Post-Bariatric Hypoglycemia in Individuals with Obesity and Type 2 Diabetes after Laparoscopic Roux-en-Y Gastric Bypass: A Prospective Cohort Study"

_biomedicines, 2024, doi:10.3390/biomedicines12081671_

Round 1

Reviewer 1 Report

Comments and Suggestions for Authors

The manuscript titled ‘Post bariatric hypoglycemia in individuals with obesity and T2DM after LRYGBP – A prospective cohort study’ by Kehagias et al. describes hypoglycemic phenomena in a cohort of patients with a resected fundus, aiming to identify a clinically valuable parameter predicting the onset of neuroglycopenia. The major limitation of this paper is the insufficient description of methods, including the diets of these patients, and misleading terminology, for example, ‘area mass.’ Figure and Table 6 are not described as presenting scientific findings but are simply pasted from the computer programs. Moreover, the significant changes do not necessarily become ‘predictive,’ particularly if this involves OGTT value only at a particular time period of 6 months analyzed in 24 patients. These overstatements should be removed, and the discussion should be aligned with the findings.

Introduction: The first paragraph states no problem, only that ‘the intriguing implicated pathophysiological alterations have paved the way.’ Therefore, the discussion of mechanisms, ‘research endeavors to shed light in the realm of MBS,’ is not meaningful without a stated problem. I recommend the authors start with the second paragraph stating the problem. However, in discussing Whipple's triad, it is more precise to refer to the "criteria" or "components" rather than "symptomatology," as the triad includes both symptoms and objective findings (such as low plasma glucose levels and relief of symptoms upon glucose administration). In scientific and medical contexts, it is standard to refer to pancreatic beta cells using the Greek letter "β," rather than ‘b’ used in this manuscript.

Methodology:

  1. ‘β-cell area mass’: How can one measure the mass of the area? It is either area (e.g., cm²) or mass (g). The method for measurement has not been described. Authors also use ‘β-cell mass area,’ for example, on page 368.
  2. Blood Samples: The sentence "blood samples were carried out during a 118 75gr OGTT preoperatively, at 6 and 12 months postoperatively" is not complete and has some issues with clarity and grammar.
  3. Hormone Levels: ‘Furthermore, the insulin and GLP-1 levels at 30 and 60 min postprandially were evaluated and correlated with PBH, as at these time points the most increased secretion is observed.’ This description is unclear with respect to the methods used for the measurement of hormones, as well as PBH, which was defined later.
  4. Table 5: ‘Gastrointestinal hormones, Glucagon and GLP-1 at baseline, 6 and 12 months.’ The title of this table does not represent its content, which are correlations’ p values. The type of performed analysis has not been described.

Discussion:

  1. Provocative Tests: ‘while after provocative tests’ what tests were those?
  2. Preexisting β-cell Function: Raverdy et al. suggested that PBH after LRYGBP occurs in patients with a higher preexisting β-cell function when insulin sensitivity is restored in relation to weight loss, which more succinctly explains our findings [22]. How do these findings relate to this study if preexisting β-cell function was not measured?
  3. Terminology: ‘Regarding the ideal diagnostic method, clinicians have been grappling to find the most proper approach, as well.’ The English and terminology editing are required.
  4. Mixed Meal Test: ‘From the provocative tests, the mixed meal test appears as a more preferred method because it resembles more an ordinary meal.’ This is the first mention of meals. The diet of the patients was not described, and their relevance to hypoglycemic episodes has not been neither measured nor discussed.

It is not clear how this final statement ‘According to our findings, postprandial insulin levels at 30 min, at 6 months might be used as a predictive factor for PBH at 12 months’ is related to the previous conclusion ‘Therefore, according to the literature and our findings, pre-existing β-cell hyperfunction, which persists after weight loss is probably the key leverage point for PBH [40,41].’ Moreover, ‘postprandial insulin levels at 30 min, at 6 months’ is a variable parameter, particularly because the diet, lifestyle, and weight have not been studied in 24 participants of this study; therefore, the connotation of this value as a ‘predictive factor’ is not justified.

Comments on the Quality of English Language

The English editing is required.

Author Response

Comment 1: “The major limitation of this paper is the insufficient description of methods, including the diets of these patients, and misleading terminology, for example, area mass”

Response 1: We really appreciate for acknowledging these severe limitations. The type of meal consumed before reporting these symptoms was not reported due to the retrospective nature of the questionnaires and the inability of patients to reliably determine this. Hence, the laboratory values with glucose below 60mg/dl were after OGTT for all patients included. We have underlined this in the patient population subsection in the methods section. Also, we have reported this as a major limitation in the last paragraph of our discussion. Lines 149-152: “All the patients postoperatively were instructed from expert dietitians to follow a diet enriched in protein with low carbohydrates. However, the exact type of meal consumed before the occurrence of these symptoms was not reported due to the retrospective nature of the questionnaire and the inability of patients to reliably determine this.” Lines 471-475: “Another major limitation of the study was the lack of data regarding the type of meal these patients have consumed, when they completed the questionnaire, since the consumption of meals with low carbohydrates and high protein is the cornerstone for avoiding PBH. Hence, the laboratory values of glucose below 60mg/dl were documented after OGTT for all the patients.”

The term “b-cell area mass” was replaced in all the text and tables with the term “β-cell mass”. Based on Meier et al. the ratio c-peptide to glucose fasting has a significant important linear regression with β-cell area (r=0.63). So, the above index was applied as “β-cell mass” in line with the above findings. This index was also used in the study from Raverdy et al. Thank you for pointing out this. We have explained also the exact relationship of this index with β-cell area in the definitions and criteria subsection.

Comment 2: “Figure and Table 6 are not described as presenting scientific findings but are simply pasted from the computer programs.”

Response 2: Thank you for your valuable comments. We have modified the table 6 heading and figure 3 legend by describing more accurate the results. Also, in section 3.5 the correlations are described more explicitly. Lines 290-293 “The high correlation coefficient between the binary glucose variable at 12 months postoperatively and the postprandial insulin at 30 and 60 min at 6 months postoperatively was an unexpected interesting finding. In order to investigate whether the post-prandial levels of insulin at 6 and 12 months could be a useful marker for PBH”. Lines 300-307 “The final step was validating the accuracy of the postprandial insulin levels at 30 min at 6 months (insulin 6 30) as a potential prognostic marker. The validation was assessed by generating ROC curves. The positive result was the occurrence of PBH. The ROC curves revealed that insulin at 30 min at 6 months is a sensitive and reliable biomarker, since its calculated AUC value was the highest at 0.933 (Figure 3). The rest of the tested biomarkers showed lower values of AUC as depicted in Figure 3. Therefore, this reinforces the potential use of insulin at 30 min at 6 months as a potential PBH prognostic biomarker.”

Comment 3: “the significant changes do not necessarily become ‘predictive,’ particularly if this involves OGTT value only at a particular time period of 6 months analyzed in 24 patients. These overstatements should be removed, and the discussion should be aligned with the findings”

Response 3: We completely agree with your thoughts and we have revised accordingly in order to mitigate our findings and do not overstate. Lines 346-347: “insulin levels at 30min during OGTT at 6 months might be used as a predictor for PBH”

Lines 405-406: “Specifically, postprandial insulin levels at 30 min at 6 months might be used as a predictive factor for PBH at 12 months, although this should be validated in a larger sample”

Lines 468-469: “…. except for the small sample size of the study cohort, which severely weakens our findings…”

Line 477-478: “…. potential predictive value…”

Line 484: “…. might be used as a predictive factor for PBH…”

Comment 4: “The first paragraph states no problem, only that ‘the intriguing implicated pathophysiological alterations have paved the way.’ Therefore, the discussion of mechanisms, ‘research endeavors to shed light in the realm of MBS,’ is not meaningful without a stated problem. I recommend the authors start with the second paragraph stating the problem”

Response 4: Thank you for your valuable suggestions. We have changed the order of the paragraphs and adjusted the references in order.

Comment 5: “in discussing Whipple's triad, it is more precise to refer to the "criteria" or "components" rather than "symptomatology," as the triad includes both symptoms and objective findings (such as low plasma glucose levels and relief of symptoms upon glucose administration)”

Response 5: Thank you for pointing this out. We have replaced symptomatology with criteria.

Comment 6: “In scientific and medical contexts, it is standard to refer to pancreatic beta cells using the Greek letter "β," rather than ‘b’ used in this manuscript”

Response 6: We absolutely agree with your indication. We have replaced everywhere b-cell with β-cell.

Comment 7: “β-cell area mass’: How can one measure the mass of the area? It is either area (e.g., cm²) or mass (g). The method for measurement has not been described. Authors also use ‘β-cell mass area,’ for example, on page 368”

Response 7: We could not agree more with your suggestion. As we stated in a previous comment, we have replaced b-cell area mass (which is wrong) with “β-cell mass”. In lines 184-187, the phrase has been adjusted by adding the units of the index “The ratio of fasting c-peptide to fasting glucose (ng/ml x mg/dl) was utilized to estimate β-cell mass. This index has been validated by Meier et al. and it has shown a significant linear correlation with histological assessment of β-cell area (r = 0.63)”

Comment 8: “Blood Samples: The sentence "blood samples were carried out during a 118 75gr OGTT preoperatively, at 6 and 12 months postoperatively" is not complete and has some issues with clarity and grammar”

Response 8: We completely agree with your suggestion. The above sentence has been rephrased as “blood samples were collected during a 75gr OGTT at 0, 30, 60 and 120 min. This procedure was repeated preoperatively, at 6 and 12 months postoperatively.”

Comment 9: “Hormone Levels: ‘Furthermore, the insulin and GLP-1 levels at 30 and 60 min postprandially were evaluated and correlated with PBH, as at these time points the most increased secretion is observed.’ This description is unclear with respect to the methods used for the measurement of hormones, as well as PBH, which was defined later”

Response 9: Thank you for this comment. We have now clarified the time points that gastrointestinal hormones were measured and also underline the rationale for particularly evaluating the postprandial levels of GLP-1 and insulin at 30 and 60min. Lines 164-165, subsection 2.3 “The aforementioned gastrointestinal (GI) hormones were all measured during the OGTT at 0, 30, 60 and 120 min, preoperatively, at 6 and at 12 months.” Also, in the next subsection lines 177-182 “Supposing that GI hormones are implicated in PBH, we particularly evaluated the postprandial levels of insulin and GLP-1 at different time points during the OGTT. Specifically, based on the literature, an abrupt postprandial increase of insulin and GLP-1 is observed at 30 and 60min postprandially. Therefore, the postprandial levels at these time points were investigated for a potential correlation with PBH.”

Comment 10: “Table 5: ‘Gastrointestinal hormones, Glucagon and GLP-1 at baseline, 6 and 12 months.’ The title of this table does not represent its content, which are correlations’ p values. The type of performed analysis has not been described”

Response 10: Sorry for that mistake and thank you for pointing this out. The title of Table 5 has changed to “Biserial correlation of parameters with the binary variable of glucose at 120 min at 12 months”. The type of analysis is now described and also in the footnotes the binary variable of glucose is explained.

Comment 11: “Provocative Tests: ‘while after provocative tests’ what tests were those?”

Response 11: Thank you for your suggestion. We have now provided the types of provocative tests. Line 354 “namely OGTT and mixed meal test”

Comment 12: “Preexisting β-cell Function: Raverdy et al. suggested that PBH after LRYGBP occurs in patients with a higher preexisting β-cell function when insulin sensitivity is restored in relation to weight loss, which more succinctly explains our findings [22]. How do these findings relate to this study if preexisting β-cell function was not measured?”

Response 12: Thank you for pointing this out. Insulinogenic index is used for evaluating β-cell function. So, we have made changes, showcasing this detail and also further elaborated on our explanation based on the mean insulinogenic index of the four patients that presented PBH. In lines 364-372 the following changes have been made: “Raverdy et al. suggested that PBH after LRYGBP occurs in patients with a higher preexisting beta-cell function, as estimated by the insulinogenic index, when insulin sensitivity is restored in relation to weight loss, which more succinctly explains our findings [22]. The mean preoperative insulinogenic index of the four patients of the cohort that presented PBH was 0.654 and it showed a remarkable 4-fold increase at 12 months reaching 2.5, which is far higher than the mean value of the insulinogenic index of the cohort, as depicted in Table 3. This indicates that after achieving weight loss and improving insulin sensitivity, β-cell function is restored and patients with an inherent increased β-cell function, will demonstrate increased values of insulinogenic index”

Comment 13: “Terminology: ‘Regarding the ideal diagnostic method, clinicians have been grappling to find the most proper approach, as well.’ The English and terminology editing are required”

Response 13: Thank you for pointing this out. We have rephrased the sentence as “The optimal diagnostic approach for the diagnosis of PBH has not yet been determined from clinicians”

Comment 14: “Mixed Meal Test: ‘From the provocative tests, the mixed meal test appears as a more preferred method because it resembles more an ordinary meal.’ This is the first mention of meals. The diet of the patients was not described, and their relevance to hypoglycemic episodes has not been neither measured nor discussed”

Response 14: We completely agree with your statement. The type of meal is absolutely implicated in the occurrence of PBH. Although, documented glucose values were after OGTT, the type of meal before the onset of symptoms when the patients completed the questionnaire was unknown. Now we have added this to the discussion and also made a comment regarding the type of meal. Lines 390-395 “Although, the documented glucose values in the studied cohort were after OGTT, the type of meal before the onset of symptoms was unknown when patients filled the questionnaires. The lack of this information was a major drawback, since it can affect the occurrence of PBH. Specifically, meals with a high-carbohydrate load are the main culprits for PBH and should be replaced with small meals, enriched in protein and fiber [1]. Finally ….”

Comment 15: “It is not clear how this final statement ‘According to our findings, postprandial insulin levels at 30 min, at 6 months might be used as a predictive factor for PBH at 12 months’ is related to the previous conclusion ‘Therefore, according to the literature and our findings, pre-existing β-cell hyperfunction, which persists after weight loss is probably the key leverage point for PBH [40,41]”

Response 15: We agree with the above comment. We have clarified in the conclusion how these two findings are related. We have add a sentence in the conclusion explaining this. In lines 484-486: “According to our findings, a pre-existing β-cell hyperfunction, which persists after weight loss postoperatively and might lead to increased postprandial insulin levels, is associated with PBH. Moreover, the….”

Comment 16: “Moreover, ‘postprandial insulin levels at 30 min, at 6 months’ is a variable parameter, particularly because the diet, lifestyle, and weight have not been studied in 24 participants of this study; therefore, the connotation of this value as a ‘predictive factor’ is not justified”

Response 16: Thank you for this comment. We need to clarify that postprandial insulin values were documented during OGTT. Furthermore, lifestyle of the included patients has not been studied. Regarding the weight loss, the four patients with PBH did not present increased weight loss compared to the patients without PBH. The overall weight change is reported in Table 1. Based on these, weight and diet were standardized, but lifestyle has not been included in our study. So we believe that insulin levels should not be considered as a variable parameter.

Comment 17: “The English editing is required”

Response 17: An English professor, working in the university, who is a native English speaker has meticulously proofread the manuscript making required changes, without changing the meaning.

Reviewer 2 Report

Comments and Suggestions for Authors

Thank the Editors for the opportunity to review this study. It a well-written manuscript about the hypoglycemia after RYGB. It showed that postprandial insulin levels at 6 months postoperatively can predict the occurrence of hypoglycemia. Nevertheless, there are some major points that have to be explained:

Material and methods

Study design - the authors showed that the primary and secondary outcomes were postprandial levels of glucose and others at 120 min after 12 month, but the results focused on the follow up of 6 months. Probably because the results are significant, but still it should be written in MM section. 

The details of surgery technique and exclusion criteria should be written in this paper. The readers won't look for it in different papers.

Results

There were two types of surgeries performed. The authors did not differentiate it in results. In my opinion the results for these types of surgeries could be different, if they are not, it should be described as well.

It should be noted if there was any T2D remission or not. The mean values of HbA1c are not enough for me as a reader.  Moreover, we don't know if the patient continued the treatment of T2D or what was the treatment before the surgery.

Discussion is nicely written. It describes all important issues regarding the pathophysiology of the topic.

Author Response

Comment 1: “Study design - the authors showed that the primary and secondary outcomes were postprandial levels of glucose and others at 120 min after 12 months, but the results focused on the follow up of 6 months. Probably because the results are significant, but still it should be written in MM section”

Response 1: Thank you for pointing this out. We have clarified this in the material and methods section. Lines 112-114 “Finally, a potential correlation between of the aforementioned parameters at 6 and 12 months, with PBH at 12 months and PBH at 12 months was also investigated.”

Comment 2: “The details of surgery technique and exclusion criteria should be written in this paper. The readers won't look for it in different papers”

Response 2: Thank you for your advice. We have now included in the patient population subsection, the exclusion criteria and the surgical technique for these two operations. In lines 119-135 “The duration of T2DM was determined less than 8 years, since longer duration is related with irreversible β-cell function impairment and less recovery postoperatively. All participants were evaluated by different specialties and behavioral, anthropometric parameters and metabolic profile were assessed. Exclusion criteria were gestation, diabetes mellitus type I, alcohol or drug abuse, major depressive disorder, non-compliance with the medical personnel’s instructions and previous abdominal surgeries with altered gastrointestinal anatomy.

All the operations were completed laparoscopically from the same surgeon at the surgical department of the tertiary referral hospital. Both procedures included the creation of a small gastric pouch of 30ml capacity, a very long biliopancreatic limb of 200cm and an alimentary limb of 150cm. The gastrojejunal anastomosis was created with a circular stapler of 25mm diameter after transoral placement of the anvil with assistance from the anesthesiologist. In the fundus resection group the gastric body and fundus were mobilized by dividing the gastrocolic ligament and short gastric vessels until exposing the angle of His. Gastric fundus with dimensions, ±5,5cm (width) and ±10cm (vertical length) was removed with the use of a linear stapler.”

Comment 3: “There were two types of surgeries performed. The authors did not differentiate it in results. In my opinion the results for these types of surgeries could be different, if they are not, it should be described as well”

Response 3: Thank you for this interesting comment. PBH was not associated with the type of surgery, since PBH occurred in patients from both groups. Also, according to the findings of our previous study fundus resection was not implicated in glucose regulation. This has been added in lines 222-223 “Finally, PBH was not associated with the type of surgery, since PBH occurred in patients from both groups.” This has also mentioned in the first paragraph of the discussion “The type of surgery was not associated with PBH, further supporting our previous findings, which showed that fundus resection is not implicated in glycemic control [8].”

Comment 4: “It should be noted if there was any T2D remission or not. The mean values of HbA1c are not enough for me as a reader.  Moreover, we don't know if the patient continued the treatment of T2D or what was the treatment before the surgery”

Response 4: Thank you for your valuable comments. We have clarified more the diabetic status of the patients and the T2DM remission by including the medications. We have made the following changes: In lines 216-222 “Regarding the preoperative diabetes treatment only one patient was totally off an-ti-diabetic medications. The other twenty-three patients were on oral medications, while seven took insulin and four GLP-1 analogs.  Postoperatively, 95% (23/24) showed complete diabetes remission at one year, without taking any medications. Only one patient from the LRYGBP with fundus resection group, experienced relapse and was re-introduced on oral medications at one year.” In Table 1 we have added the anti-diabetic medications as well.

Round 2

Reviewer 1 Report

Comments and Suggestions for Authors

After the revision, this paper has been improved but not sufficiently.

Authors are still cofused about area, mass , function, insulinogenic index,  or index of beta cells and use all of these terms in the different part of the paper as the same parameter. In the lIne 177, authors desribe this parameter as a correlation of index to and area obtained from histological slides : 'Therefore, the post- prandial levels at these time points the most increased secretion is observed. Thewere hewere investigated for a potential correlation with PBH. Finally, the ratio of fasting c-peptide to fasting glucose (ng/ml x mg/dl) was utilized to estimate β-cell mass. This index has been validated by Meier et al.., and it has shown a promising significant linear correlation with  the histological assessment of β-cell area (r = 0.63) [21, 22]'.

r=63 does not describe any significance, which is not presented.

Therefore, this parameter should be desribed as a 'calculated β-cells area' in every section of manuscript.  Area is NOT equivalent to mass and not equivalent to a function!! Each of these aspects of  β-cell biology required a different type of analysis. 

The correction of English grammar by a professor is not sufficient.

Autors should edited this manuscript for a content to improve understandidng of every sentence. In the same example :'Therefore, the post- prandial levels at these time points the most increased secretion is observed'. It remains unclear what ' levels' were measured at these time points and what molecules exhibited the most increased secretion'.

The comparison of concentrations should be shown in %  and statistical significance should be demonstrated.  Maybe this statment is based on analysis described in figure 3?  The description of resuts is neither systematic not complete.

Methods are not describing how c-peptide was measured to calculate this index.

The analysis of sentences for content should be performed in all manuscript. 

For example line 61:'The intriguing implicated pathophysiological alterations have paved the way towards relentless research, in order to elucidate the underlying mechanisms, which 62 might serve as potential pharmaceutical targets. The gastric fundus, the foregut-hindgut 63 theory, the gut microbiota, and the bile acids are only some of these recognized mecha- 64 nisms, while microRNAs have appeared recently as potential targets or promising predic- 65 tive factors [4, 8, 9]. Notwithstanding that research endeavors to shed light in the realm of 66 MBS, its increased prevalence has led clinicians to encounter more, not-well recognized 67 complications, preserving the vicious cycle of obscurity and uncertainty.'

 These statments are meaningless because it is unclrea what are 'alterations', what are mechanisms that should be studied?  What  targets are involved?

Why the listed mechanism include an organ, a theory, a 'microbiota', and a molecules 'bile acids'. Each of these  factors should be described as a mechanism. Mechanism is not a predictive factor, which is used at the end of  the sentence. This type of introduction doeas not provide any sufficient scientific context.  All text should be examined for the accuracy of terminology and context.

Comments on the Quality of English Language

The text requires extensive revision of both the content and sentence structure.

Reviewer 2 Report

Comments and Suggestions for Authors

The authors responded satisfactorily to all comments. I believe that the manuscript is worth publishing in Biomedicines.